# Clinical and Prognostic Value of Antigen-Presenting Cells with PD-L1/PD-L2 Expression in Ovarian Cancer Patients

**DOI:** 10.3390/ijms222111563

**Published:** 2021-10-26

**Authors:** Anna Pawłowska, Agnieszka Kwiatkowska, Dorota Suszczyk, Agata Chudzik, Rafał Tarkowski, Bartłomiej Barczyński, Jan Kotarski, Iwona Wertel

**Affiliations:** 1Independent Laboratory of Cancer Diagnostics and Immunology, I Chair and Department of Oncological Gynaecology and Gynaecology, Medical University of Lublin, 20-093 Lublin, Poland; agakwiatkowska2604@gmail.com (A.K.); dorotasuszczyk@umlub.pl (D.S.); agatachudzik@umlub.pl (A.C.); iwonawertel@umlub.pl (I.W.); 2I Chair and Department of Oncological Gynaecology and Gynaecology, Medical University of Lublin, 20-081 Lublin, Poland; rafaltar@yahoo.com (R.T.); bbarczynski@poczta.onet.pl (B.B.); jan.kotarski.gabinet@gmail.com (J.K.)

**Keywords:** ovarian cancer, tumor microenvironment, immune suppression, programmed cell death pathway, immunotherapy

## Abstract

The latest literature demonstrates the predominant role of the programmed cell death axis (PD-1/PD-L1/PD-L2) in ovarian cancer (OC) pathogenesis. However, data concerning this issue is ambiguous. Our research aimed to evaluate the clinical importance of PD-L1/PD-L2 expression in OC environments. We evaluated the role of PD-L1/PD-L2 in OC patients (*n* = 53). The analysis was performed via flow cytometry on myeloid (mDCs) and plasmacytoid dendritic cells (pDCs) and monocytes/macrophages (MO/MA) in peripheral blood, peritoneal fluid (PF), and tumor tissue (TT). The data were correlated with clinicopathological characteristics and prognosis of OC patients. The concentration of soluble PD-L1 (sPD-L1) and PD-1 in the plasma and PF were determined by ELISA. We established an accumulation of PD-L1^+^/PD-L2^+^ mDCs, pDCs, and MA in the tumor microenvironment. We showed an elevated level of sPD-L1 in the PF of OC patients in comparison to plasma and healthy subjects. sPD-L1 levels in PF showed a positive relationship with Ca125 concentration. Moreover, we established an association between higher sPD-L1 levels in PF and shorter survival of OC patients. An accumulation of PD-L1^+^/PD-L2^+^ mDCs, pDCs, and MA in the TT and high sPD-L1 levels in PF could represent the hallmark of immune regulation in OC patients.

## 1. Introduction

Ovarian cancer (OC) is mostly advanced during diagnosis, with widespread metastasis, and still has poor prognosis despite the new diagnostic and therapeutic approaches [1,2]. An outstanding role in the clinical outcome of patients with OC is played by the tumor microenvironment (TME). The studies prove that TME and T cell infiltration is strongly associated with the prognosis of ovarian cancer patients.

Unfortunately, the majority of OC can evade the host’s immune response and stimulate tumor development by the deactivation or death of CD8^+^ T-cells and NK cells, which play a predominant role in antitumoral immune response via immune checkpoints (ICPs) [3,4,5,6,7]. In normal conditions, the role of these structures is the regulation and maintenance of the balance between immune response and tolerance, including the inhibition of immune response targeted against host self-tissues. This is possible due to the enhancement and inhibition of the activity of T cells. One of the negative regulators of activated T cells is programmed cell death receptor 1 (PD-1) and its ligands (PD-L1, PD-L2) axis [4,8].

In pathological conditions, including cancer, the PD-1/PD-L1 axis can weaken the functioning of T cells, which suggests that PD-L1 is a pivotal biological factor included in evasion of the host’s anticancer immune response [9]. The binding of PD-L1 or PD-L2 to PD-1 on T cells leads to the inhibition of secretion of cytokines (i.e., TNF-α, IL-2, and IFN-γ) and proliferation of T cells and brings an extremely inhibitory signal, which leads to the deactivation of T cells via anergy [10,11]. T cells become exhausted, i.e., they cannot recognize and eliminate cancer cells.

The PD-1 receptor is expressed by various cells, i.e., CD4^+^ and CD8^+^ T cells, B cells. The PD-L1 expression is related to unfavorable prognosis in many tumors, including bladder cancer, non-small cell lung carcinoma (NSCLC), and kidney cancer [11,12,13,14,15,16]. Hamanishi and co-workers have proved that an increased level of PD-L1 in OC patients is related to poor overall survival (OS) and progression-free survival (PFS). Interestingly, they also showed a negative correlation between CD8^+^ T cell infiltration and the expression of PD-L1 [17]. In contrast, some authors have demonstrated an opposite correlation [18,19,20]. Moreover, the study of Maine and co-workers suggests that the PD-L1 expression on monocytes is increased in the blood of OC patients in comparison to benign tumors. These results highlight the PD-L1 expression on monocytes/macrophages (MO/MA) as a pivotal factor in TME [21]. However, the data considering prognosis based on PD-L1 is still ambiguous.

The expression of PD-L2 is not as widespread as PD-L1 and is inducible on antigen-presenting cells (APCs), such as dendritic cells (DCs) and MO/MA. Despite the 2–6-fold stronger affinity of PD-L2 than PD-L1 to PD-1, PD-L1 is the primary ligand for PD-1 and seems to be a more important prognostic and diagnostic factor [2,22,23]. PD-L2 can directly bind to the PD-1 receptor, whereas PD-L1 requires conformation changes. Therefore, in the case of the PD-L1 and PD-L2 distribution at the same level, PD-L2 binds to the PD-1 receptor. However, in normal conditions, the level of PD-L2 is significantly decreased in comparison to the level of PD-L1 [22].

Furthermore, the concentration of immunological factors in fluids, i.e., peripheral blood (PB) and peritoneal fluid (PF), may be a potential prognostic and diagnostic biomarker [9]. Soluble forms of PD-L1 (sPD-L1) are generated by metalloproteinases that cleave the membrane-bound ligand [22]. Soluble forms of ligands can bind to the PD-1 receptor in a similar way as membrane-bound ones and yield an analogous inhibitory effect. Moreover, soluble forms of both ligands can be easily detected in clinical practice by a simple blood test [24]. The determination of sPD-L1 concentrations via liquid biopsy may turn out to be a useful clinical tool in the treatment of OC patients [1,25]. Nonetheless, the role of sPD-L1 in the physiological condition remains unclear, and the data considering its functions are limited [22,25].

The aim of our research was to evaluate the PD-L1 and PD-L2 expression on antigen-presenting cells, including myeloid and plasmacytoid DCs and MO/MA, in peripheral blood, peritoneal fluid, and tumor tissue, in relation to the clinicopathological characteristics and prognosis of ovarian cancer patients. Moreover, we evaluated the concentration of the soluble form of PD-L1 and PD-1 in the peritoneal fluid and plasma of OC patients in comparison to healthy blood donors in terms of their clinical and prognostic value.

## 2. Results

### 2.1. Percentage of Myeloid (BDCA-1^+^CD19^−^) and Plasmacytoid (BDCA-2^+^CD123^+^) Dendritic Cells in Peripheral Blood, Peritoneal Fluid, and Tumor Tissue from OC Patients

The highest percentage of myeloid (BDCA-1^+^CD19^−^) DCs was detected in the peritoneal fluid of the OC patients, and it was significantly higher than in PB (*p* < 0.0001; median 1.06% vs. 0.245%), and tumor tissue (*p* < 0.001; median 1.06% vs. 0.17%), than in PB of the control group (*p* < 0.0001; median 1.06% vs. 0.225% Figure 1A).

Similarly, the highest percentage of plasmacytoid (BDCA-2^+^CD123^+^) DCs was detected in the peritoneal fluid of the OC patients, and it was significantly higher than in PB (*p* < 0.0001; median 1.03% vs. 0.215%) and tumor tissue (*p* < 0.01; median 1.03% vs. 0.15%), than in PB of the control group (*p* < 0.0001; median 1.03% vs. 0.565%; Figure 1B).

### 2.2. Percentage of Myeloid BDCA-1^+^CD19^−^ DCs with PD-L1 or PD-L2 Expression in PB, PF, and among Ovarian Cancer Infiltrating Cells

We found a significant disparity in the distribution of PD-L1 and PD-L2 positive dendritic cells in OC patients. The highest percentage of myeloid DCs with PD-L1 expression was detected among OC infiltrating cells, and it was significantly higher than in PB (*p* < 0.0001; median 55.00% vs. 17.05%) and PF (*p* < 0.05; median 55.0% vs. 41.03%; Figure 2A). Moreover, we observed a significantly higher percentage of BDCA-1^+^CD19^−^PD-L1^+^ DCs in the PB of the control group in comparison to PB of OC patients (*p* < 0.001; median 43.61% vs. 17.05%; Figure 2A).

The highest percentage of myeloid DCs with PD-L2 expression was detected in PF, and it was significantly higher than in PB of OC patients (*p* < 0.0001; median 34.50% vs. 2.94%) and in PB of the control group (*p* < 0.001; median 34.50% vs. 20.21%) and higher than in the tumor; however, the difference did not reach statistical significance (median 34.50% vs. 24.01%; *p* > 0.05; Figure 2B). Moreover, we detected a significantly higher percentage of BDCA-1^+^CD19^−^PD-L2^+^ DCs in the tumor tissue in comparison to PB of the OC patients (*p* < 0.0001; median 24.01% vs. 2.94%; Figure 2B). We also observed a higher percentage of BDCA-1^+^CD19^−^ DCs with PD-L2 in peripheral blood of the control group than in PB of OC patients (*p* < 0.001; median 20.21% vs. 2.94%).

### 2.3. Percentage of Plasmacytoid BDCA-2^+^CD123^+^ DCs with PD-L1 or PD-L2 Expression in PB, PF, and among Ovarian Cancer Infiltrating Cells

The highest percentage of plasmacytoid DCs with PD-L1 expression was detected among OC infiltrating cells, and it was significantly higher than in PB (*p* < 0.001; median 36.36% vs. 14.40%; Figure 3). We also detected a higher percentage of BDCA-2^+^CD123^+^PD-L1^+^ cells in the tumor tissue in comparison to PF; however, the difference did not reach statistical significance (*p* > 0.05; Figure 3). Moreover, we observed a higher percentage of BDCA-2^+^CD123^+^ DCs with PD-L1 in peripheral blood of the control group than in PB (*p* < 0.0001; median 63.44% vs. 14.40%), PF (*p* < 0.001; median 63.44% vs. 23.85%), and tumor tissue (*p* < 0.01; median 63.44% vs. 36.36%) of OC patients.

The highest percentage of BDCA-2^+^CD123^+^PD-L2^+^ was also detected among OC infiltrating cells, and it was higher than in PB (*p* = 0.06; median 8.53% vs. 5.40%; data not shown).

### 2.4. Percentage of Myeloid and Plasmacytoid DCs with PD-L1 or PD-L2 Expression in Patients with Different Clinical Manifestations of Ovarian Cancer

In the next step, we assessed the clinical relevance of dendritic cells with PD-L1/PD-L2 expression in OC patients. The percentage of both mDCs and pDCs expressing PD-L1/PD-L2 was correlated with specific ovarian cancer clinicopathological features, i.e., FIGO stage (I/II vs. III/IV), grading (II vs. III), and Kurman and Shih classification type (type I vs. type II). The same mDCs and pDCs expressing PD-L1/PD-L2 control were used in each subgroup analysis (Figure 4A,E,I; Figure 4B,F,J; Figure 4C,G,K; Figure 4D,H,L).

The percentage of mDCs with PD-L1 expression in peripheral blood of OC patients was lower in low (I/II) and advanced (III/IV) FIGO stages (each *p* < 0.001; Figure 4A), grade II (*p* < 0.0001) and III (*p* < 0.05; Figure 4E), and both types of OC (*p* < 0.0001 and *p* < 0.01, respectively; Figure 4I) compared to peripheral blood of healthy subjects.

The percentage of mDCs with PD-L2 expression in peripheral blood of OC patients was lower in low (I/II) and advanced (III/IV) FIGO stages (*p* < 0.001 and *p* < 0.0001, respectively; Figure 4B), grade II (*p* < 0.0001) and III (*p* < 0.001; Figure 4F), and type I (*p* < 0.0001) and type II (*p* < 0.001) of OC compared to healthy women (Figure 4J).

The percentage of pDCs with PD-L1 expression was lower in low (I/II) and advanced (III/IV) stages (both *p* < 0.0001; Figure 4C), grade II (*p* < 0.0001) and III (*p* < 0.001; Figure 4G), and type I (*p* < 0.0001) and type II (*p* < 0.001) of OC compared to healthy women (Figure 4K).

Statistical analysis revealed no significant differences in the percentage of pDCs with PD-L2 expression in the blood, PF, and TT of OC patients with different stages, grades, both Kurman–Shih types of OC compared to healthy women (*p* > 0.05; Figure 4D,H,L), (Appendix A).

We showed no significant disparity in the distribution of PD-L1 and PD-L2 positive myeloid and plasmacytoid DCs in the peripheral blood, peritoneal fluid, and tumor tissue in the different clinicopathological features of the OC patients, i.e., the FIGO stage, grade, and types of OC (*p* > 0.05).

### 2.5. Relationship between Clinicopathological Characteristics of OC Patients and the Percentage of Myeloid and Plasmacytoid DCs with PD-L1 or PD-L2 Expression in PB, PF, and among Ovarian Cancer Infiltrating Cells

There was a positive relationship between BMI of the OC patients and the percentage of BDCA-1^+^CD19^−^PD-L1^+^ cells in PF (R Spearman: 0.458; *t*(N-2) 2.364, *p* < 0.05; Figure 5A). We also observed a positive relationship that tended to be significant between BMI of OC patients and the percentage of BDCA-2^+^CD123^+^PD-L1^+^ cells in PB (R Spearman: 0.266; *t*(N-2) 1.889, *p* = 0.06; Figure 5B).

There was no significant relationship between the percentages of BDCA-1^+^CD19^−^PD-L1^+^/PD-L2^+^ or BDCA-2^+^CD123^+^PD-L1^+^/PD-L2^+^ DCs in PB, PF, and tumor tissue and the clinicopathological characteristics of the OC patients, i.e., the FIGO stage, grade, types of OC, age, menopausal status, and Ca125 concentration (*p* > 0.05).

### 2.6. Percentage of MO/MA and MO/MA with PD-L1 or PD-L2 Expression in Peripheral Blood, Peritoneal Fluid, and among Ovarian Cancer Infiltrating Cells

The highest percentage of MO/MA was detected in the peripheral blood in comparison to PF and tissue of the OC patients (*p* < 0.01 and *p* < 0.001, respectively).

The percentage of CD45^+^CD14^+^ cells with PD-L1 or PD-L2 expression was significantly higher among the OC infiltrating cells than in PB (*p* < 0.0001 and *p* < 0.01; respectively).

The percentage of CD45^+^CD14^+^PD-L2^+^ cells was also significantly higher in PF in comparison to PB of the OC patients (*p* < 0.05). The results are presented in Table 1.

### 2.7. Percentage of MO/MA with PD-L1 or PD-L2 Expression in Patients with Different Clinical Manifestations of Ovarian Cancer

Next, we assessed the clinical relevance of MO/MA with PD-L1/PD-L2 expression in OC patients. The percentage of MO/MA expressing PD-L1/PD-L2 was correlated with specific ovarian cancer clinicopathological features, i.e., FIGO stage (I/II vs. III/IV), grading (II vs. III), and Kurman and Shih classification type (type I vs. type II). The same mDCs and pDCs expressing PD-L1/PD-L2 control were used in each subgroup analysis (Figure 6A–F).

The percentage of MO/MA with PD-L1 expression in peripheral blood of ovarian cancer was lower in low (I/II) and advanced (III/IV) stages (both *p* < 0.001) (Figure 6A), grade II (*p* < 0.0001) and grade III (*p* < 0.001; Figure 6B), and type I (*p* < 0.0001) and type II of OC (*p* < 0.01; Figure 6C) compared to healthy women.

The percentage of MO/MA with PD-L2 expression was lower in low (I/II) and advanced (III/IV) FIGO stages (both *p* < 0.0001) (Figure 6D), grade II and III (both *p* < 0.0001; Figure 6E), and both types of OC (both *p* < 0.0001; Figure 6F) compared to healthy women.

The percentages of CD45^+^CD14^+^ cells with PD-L1 or PD-L2 expression in PB, PF, and TT did not differ significantly (*p* > 0.05) between the different FIGO stages, grades, or Kurman–Shih types of OC (*p* > 0.05; Appendix A).

We found a significantly higher percentage of MO/MA with the expression of PD-L1 in the peripheral blood of the healthy blood donors in comparison to PB (median 99.82% vs. 10.72%; *p* < 0.0001; Figure 7A), PF (median 99.82% vs. 20.83%; *p* < 0.001), and TT (median 99.82% vs. 40.00%; *p* < 0.0001) of OC patients. Additionally, we found that the percentage of MO/MA with PD-L1 expression in the PF was significantly higher than in the PB of OC patients (median 20.83% vs. 10.72%; *p* < 0.05).

Moreover, we observed differences in the distribution of PD-L2 positive MO/MA in OC. We found a significantly higher percentage of MO/MA with the expression of PD-L1 in the peripheral blood of the healthy blood donors in comparison to PB (median 71.34% vs. 1.2%; *p* < 0.0001), PF (median 71.34% vs. 5.73%; *p* < 0.0001), and TT (median 71.34% vs. 7.58%; *p* < 0.0001) of OC patients (Figure 7B). Additionally, we found that the percentage of MO/MA with PD-L2 expression was significantly higher in PF than in PB of ovarian cancer patients (median 5.78% vs. 1.2%; *p* < 0.01).

There was no relationship between the percentage of PD-L1 and PD-L2 positive MO/MA in PB, PF, and tumor tissue, and the FIGO stage, grade, type of OC, age, menopausal status, BMI, and Ca125 level (*p* > 0.05).

### 2.8. Concentration of sPD-L1 and sPD-1 in Patients with Ovarian Cancer and Control Group

The concentrations of sPD-L1 in the plasma of healthy subjects and the plasma and PF of the OC patients are presented in Figure 8A.

The level of sPD-L1 in the PF was significantly higher than in the plasma of OC patients (median 100.5 pg/mL vs. 63.37 pg/mL; *p* < 0.001) and plasma of healthy subjects (median 100.5 pg/mL vs. 60.40 pg/mL; *p* > 0.01). The plasma sPD-L1 concentrations did not differ significantly between OC patients and control group (median 63.37 pg/mL vs. 60.40 pg/mL; *p* > 0.05).

### 2.9. Concentration of sPD-L1 in the Plasma and Peritoneal Fluid of OC Patients with Different FIGO Stages, Grades, and Kurman–Shih Types of Ovarian Cancer

The plasma and PF sPD-L1 levels did not differ significantly (*p* > 0.05) between the different FIGO stages, grades, or Kurman–Shih types of OC. We did not observe a statistically significant difference between the levels of sPD-L1 in plasma of OC patients regardless of the clinicopathologic characteristics compared with the control group.

The associated plasma and peritoneal fluid sPD-L1 levels are presented in Table 2.

### 2.10. Assessment of the Relationship between the sPD-L1 Level in Plasma and PF and the Clinical Data of OC Patients

There was a positive relationship between the sPD-L1 level in PF and the Ca125 concentration in the plasma of the OC patients (R Spearman: 0.471; *t*(N-2) 2.875; *p* < 0.01; Figure 9).

There was no significant correlation between the PF sPD-L1 concentration and the FIGO stage, grade, Kurman–Shih type, BMI, and menopausal status of the ovarian cancer patients (*p* > 0.05).

### 2.11. Relationship between the Level of sPD-1 and sPD-L1 in Plasma of OC Patients and Control Group

In our previous study, we established the sPD-1 level in the plasma and PF of ovarian cancer patients and its correlations with clinical data and patient outcomes [26].

In the present study, we found a positive relationship between the sPD-1 and sPD-L1 level in the plasma of the control group (R Spearman: 0.673; *t*(N-2) 2.57; *p* < 0.05; data not shown). There was no significant correlation between sPD-1 and sPD-L1 levels in the plasma and PF of ovarian cancer patients (*p* > 0.05).

The concentrations of sPD-1 (pg/mL) and sPD-L1 (pg/mL) in the plasma and PF of the OC patients and the plasma of the healthy donors are presented in Table 3.

### 2.12. Prediction of OC Patient Prognosis Based on Immune Parameters

We performed examinations to establish OS estimates as a function of the PD-L1 and PD-L2 expression on mDCs, pDCs, and MO/MA in PB, PF, and tumor tissue from ovarian cancer patients.

There was no significant correlation between the percentage of mDCs, pDCs, and MO/MA with PD-L1 and PD-L2 expression in PB and OC patient survival (*p* > 0.05).

We found that OC patients with a higher percentage of PD-L2 positive macrophages in PF had longer OS than those with a lower percentage of these cell populations (median 13 vs. 32 months; *p* < 0.001; Figure 10A).

There was no significant correlation between the percentage of mDCs, pDCs, and MO/MA with PD-L1 and PD-L2 expression in PF and OC patient survival (*p* > 0.05).

We also found that OC patients with a higher percentage of PD-L1 positive macrophages in tumor tissue had longer OS than those with a lower percentage of these cell populations (median 44.5 vs. 39 months; *p* < 0.05; Figure 10B).

Moreover, the obtained results implicated that OC patients with a higher percentage of plasmacytoid BDCA-2^+^CD123^+^PD-L2^+^ DCs in the tumor tissue had longer OS than patients with a lower percentage of these cell subpopulations (median 42 vs. 43 months; *p* < 0.01; Figure 10C).

There was no other significant correlation between the percentage of mDCs, pDCs, and MO/MA with PD-L1 expression among ovarian cancer infiltrating cells and the OC patient survival (*p* > 0.05).

### 2.13. Soluble PD-L1 (sPD-L1) and PD-1 (sPD-1) as a Potential Prognosis Biomarker for OC Patients

In our previous report, we found that OC patients with a lower soluble PD-1 level in the plasma, but not in PF (*p* > 0.05), had beneficial 5-year survival than those with a higher sPD-1 concentration [26].

In the present study, we found that OC patients with a higher level of sPD-L1 in the PF had shorter 5-year survival than those with a lower sPD-L1 concentration (median 48 vs. 27 months; Figure 11). There was no significant difference in the OS of the OC patients in relation to the sPD-L1 level in plasma (*p* > 0.05).

### 2.14. Prediction of Patient Prognosis Based on Clinical Characteristics of OC Patients

We performed examinations to establish OS estimates as a function of clinical parameters of ovarian cancer patients, including the FIGO stages, grade, type of OC according to Kurman and Shih, menopausal status, and BMI.

Interestingly, we demonstrated that OC patients with early stages of the disease (I and II FIGO stages) had longer OS than those with advanced disease (III and IV FIGO stages) (median 59 vs. 29 months, *p* < 0.001; Figure 12A).

We found that OC patients with grade 2 had beneficial OS in comparison to those with grade 3 (median 41.5 vs. 27 months; *p* < 0.05; Figure 12B).

We also found that OC patients before menopause had beneficial OS in comparison to patients after menopause (median 52 vs. 31 months; *p* < 0.05; Figure 12C).

We did not demonstrate statistically significant differences between the OS of OC patients with I and II type of tumor according to Kurman and Shih (*p* > 0.05).

## 3. Discussion

Despite the implementation of new biological drugs, such as Olaparib and Bevacizumab, in clinical practice, the prognosis for OC patients is still devastating. Numerous scientists are focused on the development of effective treatment of ovarian cancer. Immunotherapies based on PD-1/PD-L1/PD-L2 signaling inhibition seem to be promising therapeutical tools in the OC treatment. However, the response of OC patients to implementation of PD-1 and PD-L1 inhibitors is disappointing and totals approximately 6–22%.

It should be emphasized that in the OC microenvironment, including stromal tissue and cancer cells, two PD-1 ligands, i.e., PD-L1 and PD-L2, are expressed, and both allow cancer cells to evade host immune response via T cell exhaustion.

PD-L2 is still unexplored in terms of ICP blockade and clinical implications, which is connected with its low expression and lower impact on TME. Interestingly, the increased PD-L2 expression is detectable in ovarian cancer tissue and stroma but not in normal ovary tissue. These findings show that PD-L2 expression is inducible via the OC microenvironment during the progression of the disease. Thus, it is necessary to investigate the role of not only PD-1 and PD-L1 but also PD-L2 to develop the most effective approaches aimed at the maximal blockade of programmed cell death pathway signaling [27].

In our study, we evaluated the expression of PD-L1 and PD-L2 on antigen-presenting cells, e.g., myeloid and plasmacytoid DCs and monocytes/macrophages in the three different OC environments, i.e., peripheral blood, peritoneal fluid, and among ovarian cancer infiltrating cells (tumor tissue). In addition, we evaluated the level of sPD-L1 and sPD-1 in the plasma and PF of ovarian cancer patients. The data were integrated with different clinical characteristics and 5-year survival of the OC patients. To the best of our knowledge, this is the first study determining the expression of PD-L1 and PD-L2 on different DC subsets in ovarian cancer patients by flow cytometry.

Interestingly, we demonstrated significant differences in the distribution of BDCA-1^+^CD19^−^ myeloid (mDCs) and BDCA-2^+^CD123^+^ plasmacytoid (pDCs) DCs and these cell subpopulations with PD-L1/PD-L2 expression in OC patients. We established the highest percentage of mDCs and pDCs with PD-L1 expression among ovarian cancer infiltrating cells in comparison to PB and PF of the OC patients. Moreover, we reported the accumulation of mDCs and pDCs with PD-L2 expression among ovarian cancer infiltrating cells.

The different levels of the PD-L1 and PD-L2 expression on DCs in the three different OC environments support the existence of immunological heterogeneity in ovarian cancer [28].

It should be stressed that DCs provide a link between innate and adaptive immune response via presenting antigens to T cells and enhancing immune response and are key drivers of the anticancer immune response. They play a predominant role in the antitumor immune response via both priming and activation of T cells. However, during cancer progression, DCs are able to switch their immunostimulatory activity into the immunosuppressive one, which enhances cancer progression. Nevertheless, the mechanism of the transition is still unknown [29]. Despite the observation that macrophages are the main source of PD-L1, it has been proven that PD-L1 expression on dendritic cells plays a predominant role in the control and regulation of antitumor immune response. Interestingly, the PD-L1 deletion on DCs leads to the stimulation of antitumor response of CD8^+^ T cells. In turn, the deletion of PD-L1 from macrophages does not enhance the response [30].

In the light of the above-mentioned data, DCs are a subset of promising immune system cells that can be a powerful tool in active immunotherapy based on directing the immune system of the host to specific tumor-associated antigens [31].

We also aimed to establish the dependence between mDCs and pDCs with PD-L1/PD-L2 expression and the clinicopathological features of the OC patients. Our results did not demonstrate relationships between the PD-L1 expression on mDCs and pDCs and the clinicopathological characteristics (including the FIGO stage, grade, and type of OC) and the clinical data of the OC patients (i.e., age, Ca125 concentration, menopausal status). Similarly, using immunohistochemistry, Nhokaew and co-workers demonstrated no relationships between the PD-L1 expression and the age or menopausal status and no differences in the levels of PD-L1 expression in type I and II of OC [32].

However, it is worth highlighting the relationships between the percentage of mDCs with PD-L1 expression in ascites and BMI of OC patients. The impact of such clinical data as age, menopausal status, and BMI is not well established in the context of the PD-L1 expression in ovarian cancer. Interestingly, obesity has a paradoxical impact on cancer development [32]. It is well established that obesity inhibits the action of the immune system and is a risk factor for the development of many cancers, including ovarian cancer [33,34]. Furthermore, there is a relationship between a high level of inflammatory and hormonal factors, which may lead to tumor development [35]. Increased BMI is also related to a higher production of proinflammatory cytokines, leptin, fatty acids, and insulin. All these factors enhance the initiation and progression of cancer. Interestingly, Wang and co-workers demonstrated that obesity (BMI > 30) inhibits the activity of T cells. On the other hand, increased BMI enhances the efficiency of immunotherapies based on anti-PD-1/PD-L1 mAbs. Probably, this is related to the activation of the PD-1/PD-L1 targets via unknown mechanisms in obese patients [33].

It should be stressed that the level of PD-L1 expression on particular APC populations varies. Oh and co-workers reported that macrophages are the main source of PD-L1 in both human and mouse tumors [30].

In our study, we demonstrated the accumulation of MO/MA with PD-L1 or PD-L2 expression in the tumor tissue in comparison to PB and PF of the OC patients. Similarly, Xue et al. demonstrated elevated expression of PD-L1 and PD-L2 in tumor tissue of OC patients [36]. In addition, Gottlieb and co-workers showed the expression of PD-L1 on tumor-associated macrophages (TAMs), especially in primary OC and high-grade serous ovarian cancer (HGSOC). The expression of PD-L1 on TAMs was estimated at 74%, whereas the expression of PD-L1 on ovarian cancer cells was only 7% [37].

In our cohort of patients, we did not find a correlation between the percentage of PD-L1 and PD-L2 positive MO/MA and the clinicopathological factors of the OC patients. It is well known that some clinical features, including the FIGO stage, grade, type of OC, menopausal status, and Ca125 concentration, are prognostic factors for OC patients. However, the relationship between the percentage of MO/MA with PD-L1/PD-L2 expression and the clinical manifestation of OC is still ambiguous. Similarly, Hamanishi et al. did not observe a correlation between the PD-L1 expression and some clinicopathological features, such as the histological type and clinical stage of OC [17]. In contrast, Xue et al. showed that the expression of PD-L1 and PD-L2 on ovarian cancer cells is strongly associated with the FIGO stage of OC, which is the most important prognostic factor for OC patients [36]. Therefore, further studies are needed to verify these disparities.

MO/MA are a population of highly plastic immune system cells with the immunophenotype dependent on the impact of the microenvironment. It should be emphasized that the expression of PD-L1 and PD-L2 on MO/MA is dynamic and highly dependent on cytokines in TME. The OC microenvironment may influence MA, and they can display a protumoral and immunosuppressive immunophenotype [38,39]. The pivotal role of TAMs in ovarian cancer development is the induction of immunotolerance [40]. However, it should be emphasized that the role of PD-L2 in OC pathogenesis remains unclear and it is not as extensively explored by scientists as PD-1 and PD-L1. One of the reasons may be the fact that the level of PD-L2 expression is lower than that of PD-1 and PD-L1 [38]. There are many factors inducing PD-L2 expression on MO/MA, including IFN-α, CSF-1, and IL-4 [40].

In our study, we performed examinations to establish overall survival estimates as a function of the PD-L1 and PD-L2 expression on mDCs, pDCs, and MO/MA in the PB, PF, and tumor tissue of OC patients.

The data of the diagnosis and predicting survival based on PD-L1 and PD-L2 are still ambiguous. It should be stressed that, in the majority of scientific studies, the PD-L1 expression in tumor tissue was evaluated by immunohistochemistry, and only a few studies analyzed the expression of PD-L1 and PD-L2 on APCs by flow cytometry.

Our results did not show a relationship between the survival of ovarian cancer patients and the PD-L1 and PD-L2 expression on particular DC subsets and MO/MA in peripheral blood. It is interesting that our results for the first time implicated that OC patients with a higher percentage of PD-L2 positive macrophages in the PF had longer OS than those with a lower percentage of these cells.

Moreover, our study also demonstrated that an elevated percentage of macrophages with PD-L1 expression as well as pDCs with PD-L2 expression in tumor tissue is related to improved survival of ovarian cancer patients.

Dissimilarly, a higher PD-L2 expression in tissue in hepatocellular carcinoma and renal cell carcinoma is correlated with unfavorable prognosis, and there is evidence that increased PD-L2 expression may promote metastasis in these patients [41]. Zhang et al. found that pancreatic ductal adenocarcinoma (PDAC) patients with increased PD-L2 expression had reduced OS in comparison to those with a lower PD-L2 level in the tumor tissue [42]. Moreover, Takamori et al. reported no relationship between PD-L2 expression in tumor tissue and clinicopathological factors and survival in non-small cell lung cancer patients [43].

Interestingly, Xue and co-authors showed shorter survival of OC patients in the group with high PD-L1 expression in tumor tissue [36]. Some authors [44,45,46,47,48,49] have reported a correlation of increased PD-L1 levels in the tumor tissue of OC patients with worse outcomes and the late stage of the disease.

Yasuoka and colleagues demonstrated that PD-L2 expression on monocytes in peripheral blood was related to unfavorable prognosis in hepatocellular carcinoma (HCC) patients [50]. It should be highlighted that, as shown by Hamanishi and co-workers, high PD-L2 expression in tumor tissue is correlated with a reduced 5-year survival rate in OC patients in comparison to OC patients with a lower PD-L2 level [17]. Nevertheless, the differences did not reach statistical significance, and the method used in that study, i.e., immunohistochemistry, was different from that employed in our study (flow cytometry).

It is well established that PD-L1 is a negative regulator of anticancer response. However, very little is known about both membranous and cytoplasmatic PD-L1 expression on APCs in OC, and there are very scarce studies on soluble PD-L1 as a predicting and prognostic factor in OC patients [9]. It is still controversial whether sPD-L1 influences T cell functions.

In our research, we evaluated and compared the sPD-L1 level in the plasma and PF of the OC patients and in plasma of healthy subjects to establish the source of immunosuppression in the OC patients.

Interestingly, we showed elevated levels of sPD-L1 in PF than in the plasma of the OC patients. The concentration of sPD-L1 in PF was nearly 1.6 times higher in comparison to that in the plasma of the OC patients.

The accumulation of sPD-L1 observed in our study group in ascites may stimulate metastasis of OC and augment the immunosuppression in the peritoneal cavity. There is evidence that high levels of sPD-L1 may have an influence on metastasis in various types of malignancy, including ovarian cancer. It is well known that ascites are a way of metastasis in OC. The most frequent mechanism consists of the release of cancer cells from the primary tumor to the peritoneal cavity. Abiko et al. were the first to suggest that there is a correlation between cancer cell dissemination in the peritoneal cavity and PD-L1 expression in ovarian cancer. In a murine model of OC, they showed a significant correlation between PD-L1 expression in tumor cells and the promotion of cancer cell dissemination in the peritoneal cavity. Zhang et al. demonstrated that a high sPD-L1 level in the serum of NSCLC patients is significantly correlated with abdominal organ metastasis [51]. Moreover, Asanuma et al. proved that there is a significant correlation between the sPD-L1 level in serum and OS and the metastasis-free survival in soft tissue sarcoma (STS). The study indicates that high sPD-L1 concentration in serum is related to poor survival and metastasis in STS patients [52]. In contrast, Zheng and co-workers showed that a high concentration of sPD-L1 in the serum of gastric cancer patients is associated with the absence of lymph node metastasis [53].

Interestingly, in our study, we observed a positive relationship between levels of sPD-L1 in ascites and Ca125 in plasma of ovarian cancer patients. It should be emphasized that the concentration of Ca125 as a single factor is an unreliable OC biomarker, as its level is increased only in 50% of early cases of OC [1]. Ca125 is not an ideal marker in OC diagnostics because of its low specificity (less than 90%) and sensitivity (approximately 70%). However, it is widely used in clinical practice because of the lack of better indicators. The level of Ca125 in serum is used in the detection of ovarian cancer and monitoring the advancement of the disease. Interestingly, recent studies report that the binding mesothelin to Ca125 plays a crucial role in metastasis creation via spreading ovarian cancer cells in the peritoneal cavity [54,55]. Numerous studies reported increased concentration of Ca125 in ovarian cancer patients enhances the motility and invasiveness of cancer cells that lead to metastasis related to metastasis creation and disease progression [56,57,58].

Therapy targeted at PD-L1 and sPD-L1 may be an up-and-coming tool in the treatment and prevention of peritoneal dissemination [59]. It has been established that the level of sPD-L1 is positively correlated with the tumor size and inhibition of cytokine production, T cells proliferation, and CD8^+^ cytotoxicity [18,25]. These findings indicate that the soluble form of PD-L1 may play a role in the evasion of the immune system response by cancer cells.

Our results demonstrated that the sPD-L1 level was independent of various clinical characteristics of the OC patients (including the FIGO stage, grade, and type of OC) and clinical data of the OC patients (i.e., age, menopause status, BMI).

It should be highlighted that the elevated levels of sPD-L1 in the PF of the OC patients versus plasma and the control group led us to hypothesize that sPD-L1 may lead to local immune suppression [1,52]. Interestingly, Frigola and co-workers demonstrated that activated DCs are the source of sPD-L1. What is interesting is that in our earlier study we showed accumulation of myeloid and plasmacytoid DCs in the peritoneal fluid of OC patients [60]. These findings indicate that sPD-L1 is released by both the immune system and cancer cells [61].

Interestingly, we demonstrated that the increased sPD-L1 concentration in PF was related to the shorter survival of patients with OC. The negative impact of a high sPD-L1 level was confirmed in other solid tumors, including renal, gastrointestinal, lung cancer, and soft tissue sarcomas [1,52,61,62]. The elevated levels of sPD-L1 in ascites of our cohort of patients may lead to strong immunosuppression, disease progression, and metastasis [63,64].

In our earlier study, we demonstrated a correlation between the level of sPD-1 in plasma and PF and the survival of OC patients. Our results indicate that a significantly lower concentration of sPD-1 in the plasma is associated with beneficial OS of OC patients [26]. In contrast, Elhang and co-workers demonstrated that a higher sPD-1 level is related to beneficial survival in a murine model of OC via reduction in immunosuppression. The soluble form of PD-1 binds to PD-L1, which prevents the binding of PD-L1 with membrane-bound PD-1 on T cells. Thus, both sPD-1 and sPD-L1 are promising factors for the development of target therapies in various types of malignancies, e.g., ovarian cancer [65].

## 4. Materials and Methods

### 4.1. Patients, Ethics Statement, Standard Protocol Approvals

Our study involved 76 patients, including 23 healthy blood donors and 53 ovarian cancer patients that had undergone radical treatment at the I Chair and Department of Oncological Gynaecology and Gynaecology (Independent Public Clinical Hospital No. 1, Medical University of Lublin, Poland). The clinical characteristics of the OC patients are presented in Table 4. The diagnosis of OC was confirmed by the histopathological reports. The tumors were classified and graded according to the Silverberg grading system by two independent gynecological pathologists [66]. All the tumors were staged according to the International Federation of Gynecologists and Obstetricians (FIGO classification) [67]. The tumor type was determined according to the Kurman and Shih classification [2,68,69]. The exclusion criteria for the study cohort included a history of previous malignancies, chemotherapy, or radiation therapy prior to surgery as well as allergic, autoimmune, and infectious diseases.

All patients received standard adjuvant chemotherapy with carboplatin and paclitaxel. FIGO III and IV patients who were sub optimally debulked (residual disease > 1 cm) were additionally treated with bevacizumab. Subjects who progressed or developed recurrence were next administered second-line chemotherapy according to local standards. Data on the overall survival of the patients were obtained from the Document Personalization Center of the Ministry of Internal Affairs and Administration, Department of Protection of Confidential Information in Warsaw.

Written informed consent was obtained from all OC patients. The research received approval from the Bioethics Committee at the Medical University of Lublin (KE-0254/280/2015; KE-0254/296/2017).

The control group of the study included 23 healthy blood donors between the ages of 34 and 64 (median 57 years). The peripheral blood of the healthy donors was obtained from the Regional Centre of Blood Donation and Blood Treatment in Lublin. Every patient signed a written consent to participate in the study, and the experiments were conducted in accordance with the Declaration of Helsinki. Table 4. summarizes the clinical characteristics of the OC patients.

### 4.2. Isolation of Mononuclear Cells (MNCs)

All peripheral blood samples were collected in heparinized tubes (Sarstedt, Germany) before the surgical procedure and processed immediately. Mononuclear cells (MNCs) were separated by density gradient centrifugation at 700× *g* with a Gradisol L (AquaMedica, Poland) for 20 min at room temperature. Then, interphase cells were collected, washed twice, and resuspended in phosphate-buffered saline (PBS, PAA Laboratories GmbH, Austria).

Peritoneal fluid (PF) and tumor tissue (TT) were collected aseptically during the operation. PF mononuclear cells were isolated by density gradient centrifugation at 700× *g* on a Gradisol L (AquaMedica, Poland) for 20 min at room temperature. The subsequent procedure was the same as for MNCs isolated from PB.

Freshly resected ovarian cancer tumor tissue was minced and placed into a gentleMACS C tube and then processed with a Tumor Dissociation Kit (Miltenyi Biotec) to isolate tumor infiltrating MNCs. The obtained cell suspensions were filtered through a mesh filter (70 mm; BD Biosciences), which was followed by density centrifugation at 700× *g* on a Gradisol L (AquaMedica, Poland) for 20 min at room temperature. Afterward, cells from the interphase were collected, washed twice in phosphate-buffered saline (PBS, PAA Laboratories GmbH, Austria), and resuspended in PBS [26,70].

The viability of MNCs was evaluated by trypan blue staining; the viability was always >95%. The quantification of viable cells was performed in a Neubauer chamber.

### 4.3. Flow Cytometric Analysis

Mononuclear cells (1 × 10^6^ cells) isolated previously from PB, PF, and tumor tissue were incubated with fluorochrome-labeled monoclonal antibodies (mAbs) against cell-surface markers: anti-BDCA-1 FITC (MACS Miltenyi), anti-CD19 PerCP-Cy5.5 (BD Pharmingen), anti-BDCA-2 FITC (MACS Miltenyi), anti-CD123 PE-Cy7 (Biolegend), anti-CD45 FITC (BD Pharmingen), anti-CD14 PE-Cy7 (BD Pharmingen), anti-PD-L1 APC (Biolegend), and anti-PD-L2 PE (Biolegend) for 20 min at room temperature. Then, the cells were washed twice with PBS, and the percentage of myeloid BDCA-1^+^CD19^−^ DCs and plasmacytoid BDCA-2^+^CD123^+^ DCs, and CD45^+^CD14^+^ MO/MA with PD-L1 or PD-L2 expression was analyzed using flow cytometry (FACSCanto I Becton Dickinson, USA). The frequency of mDCs, pDCs, and MO/MA are presented as the percentage of mononuclear cells. For each tube, 100,000 events were acquired and analyzed using FacsDiva software. The expression levels of PD-L1/PD-L2 are presented as the percentage of total respective cell subsets (i.e., myeloid BDCA-1^+^CD19^−^, plasmacytoid BDCA-2^+^CD123^+^ DCs, and CD45^+^CD14^+^ MO/MA). The method of identification pDCs with PD-L1/PD-L2 expression is presented in Figure 13.

### 4.4. ELISA

Soluble PD-L1 (sPD-L1) and PD-1 (sPD-1) levels in the plasma and PF of the OC patients and in the plasma of the healthy blood donors were investigated via an immunoassay kit ELISA (sPD-L1 R&D Systems, Minneapolis, MN, USA; detection range 25 to 1600 pg/mL; sensitivity 4.52 pg/mL; and sPD-1 Biorbyt LLC., San Francisco, California, USA; detection range 15.6 to 1000 pg/mL; sensitivity 3.9 pg/mL) as specified by the manufacturers. Plate absorbance was read on an ELX-800 plate reader (BioTek Instruments, Inc, USA) and analyzed by Gen5_ (BioTek Instruments, Inc). The concentrations of sPD-L1 and sPD-1 were calculated via interpolation from a standard curve.

### 4.5. Statistical Analysis

The statistical analysis of the results was conducted using Statistica 13.0 PL and GraphPad Prism 9.

The Wilcoxon paired test was applied to compare the results obtained from the peripheral blood, peritoneal fluid, and tumor tissue of the same OC patients. The Mann–Whitney *U* test was used for comparison of the tested groups, i.e., the group of the OC patients and the control (healthy blood donors). The Spearman rank correlation test was applied to investigate the relationship between two parameters. The obtained data are presented as the median, minimum, and maximum.

The Kaplan–Meier method was applied to estimate the probability of overall survival. The log-rank test was used to calculate the differences in the survival curves. A *p*-value less than 0.05 (*p* < 0.05) was considered statistically significant.

## 5. Conclusions

In our study, we showed differences in the distribution of myeloid and plasmacytoid DCs and MO/MA with PD-L1/PD-L2 expression in OC patients. We reported that the accumulation of PD-L2 positive macrophages in ascites was associated with improved survival of our cohort of patients.

Moreover, we established an accumulation of PD-L1/PD-L2 expressing mDCs, pDCs, and MA in the TME, which may augment immunosuppression in OC patients.

Additionally, we showed an elevated level of sPD-L1 in the PF of OC patients in comparison to plasma and healthy subjects. Moreover, the sPD-L1 level in PF showed a positive relationship with Ca125 concentration. We have established an association between a higher sPD-L1 level in PF and shorter survival of OC patients.

Our results led us to conclude that the accumulation of PD-L1/PD-L2 expressing mDCs, pDCs, and MA in the TME and high sPD-L1 production in the peritoneal cavity could represent the hallmark of immune regulation in OC patients.

## Figures and Tables

**Figure 1 ijms-22-11563-f001:**
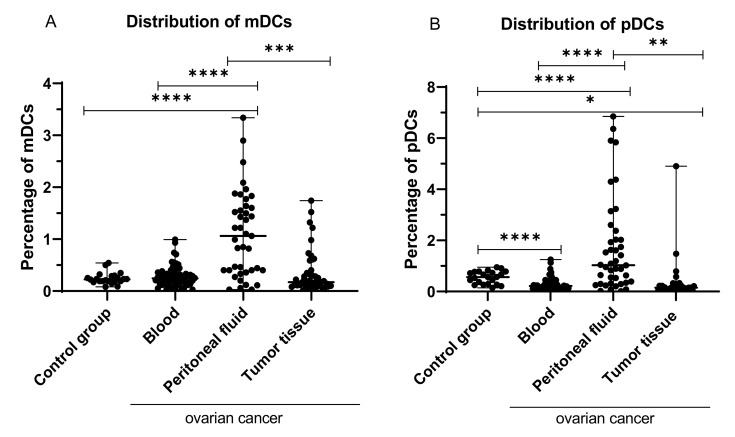
Percentage of BDCA-1^+^CD19^−^ (**A**) and BDCA-2^+^CD123^+^ cells (**B**) in healthy blood donors and peripheral blood, peritoneal fluid, and among ovarian cancer infiltrating cells. The median value with the following marks indicate statistically significant differences: * *p* < 0.05, ** *p* < 0.01, *** *p* < 0.001, **** *p* < 0.0001.

**Figure 2 ijms-22-11563-f002:**
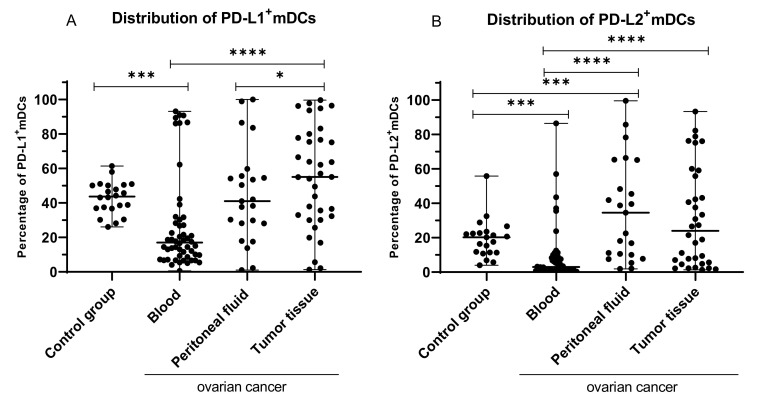
Percentage of BDCA-1^+^CD19^−^ cells with PD-L1 (**A**) and PD-L2^+^ expression (**B**) in healthy blood donors and peripheral blood, peritoneal fluid, and among ovarian cancer infiltrating cells. The median value with the following marks indicate statistically significant differences: * *p* < 0.05, *** *p* < 0.001, **** *p* < 0.0001.

**Figure 3 ijms-22-11563-f003:**
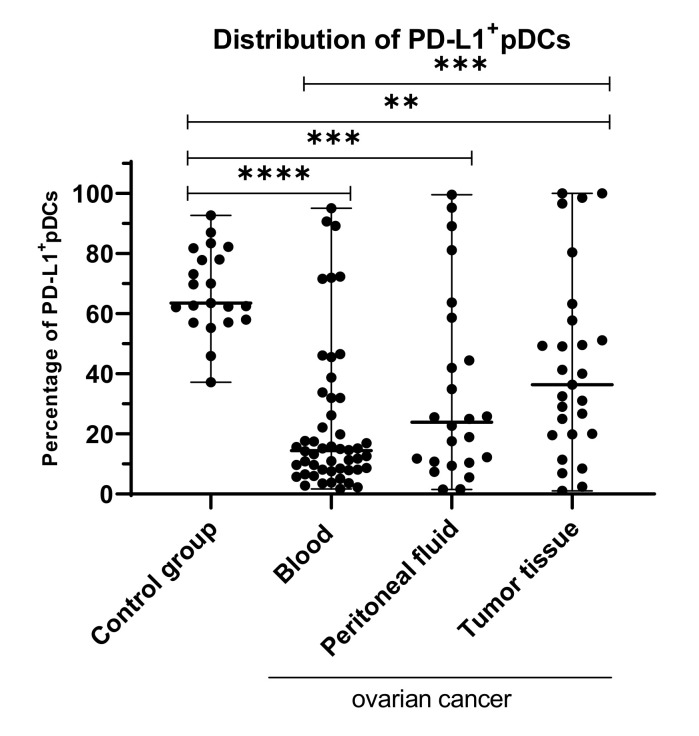
Percentage of BDCA-2^+^CD123^+^PD-L1^+^ cells in healthy blood donors and peripheral blood, peritoneal fluid, and among ovarian cancer infiltrating cells. The median value with the following marks indicate statistically significant differences: ** *p* < 0.01, *** *p* < 0.001, **** *p* < 0.0001.

**Figure 4 ijms-22-11563-f004:**
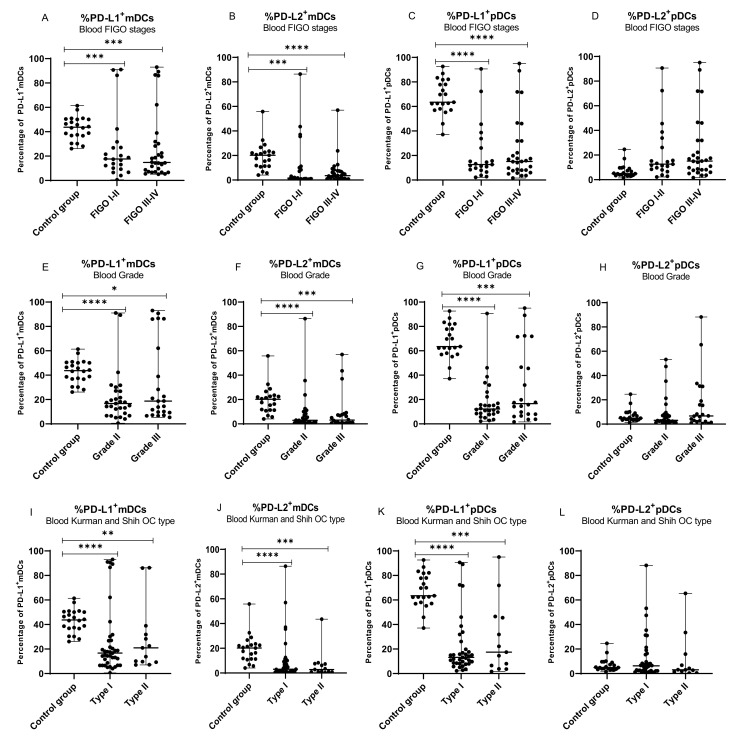
Distribution of mDCs and pDCs with PD-L1 and PD-L2 expression in peripheral blood of healthy blood donors and peripheral blood of ovarian cancer patients with different FIGO stages (**A–D**), grade (**E–H**), and OC types (**I–L**). The median value with the following marks indicate statistically significant differences: * *p* < 0.05, ** *p* < 0.01, *** *p* < 0.001, **** *p* < 0.0001.

**Figure 5 ijms-22-11563-f005:**
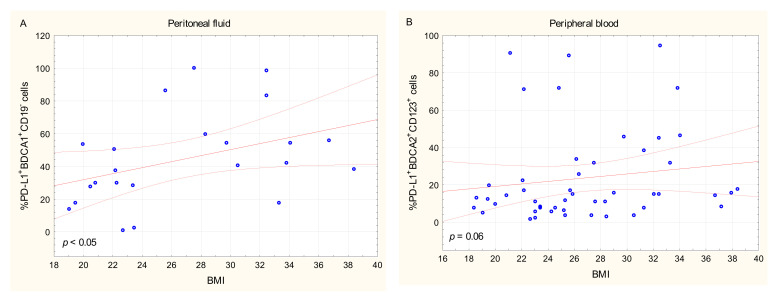
Relationship between BMI of OC patients and the percentage of BDCA-1^+^CD19^−^PD-L1^+^ cells in peritoneal fluid (**A**) and BDCA-2^+^CD123^+^PD-L1^+^ cells in peripheral blood (**B**).

**Figure 6 ijms-22-11563-f006:**
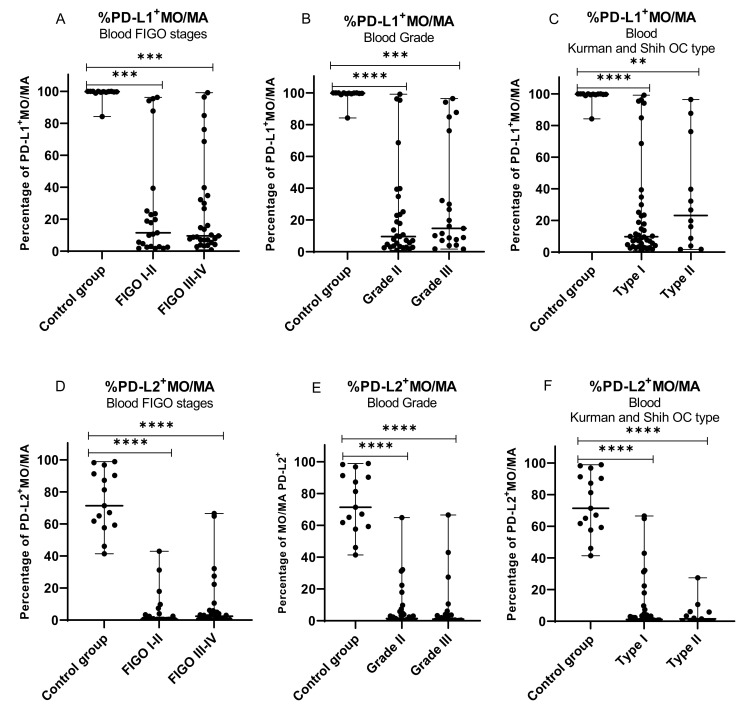
Distribution of MO/MA with PD-L1 and PD-L2 expression in peripheral blood of healthy blood donors and peripheral blood of ovarian cancer patients with different FIGO stages (**A**,**D**), grade (**B**,**E**), and OC types (**C**,**F**). The median value with the following marks indicate statistically significant differences: ** *p* < 0.01, *** *p* < 0.001, **** *p* < 0.0001.

**Figure 7 ijms-22-11563-f007:**
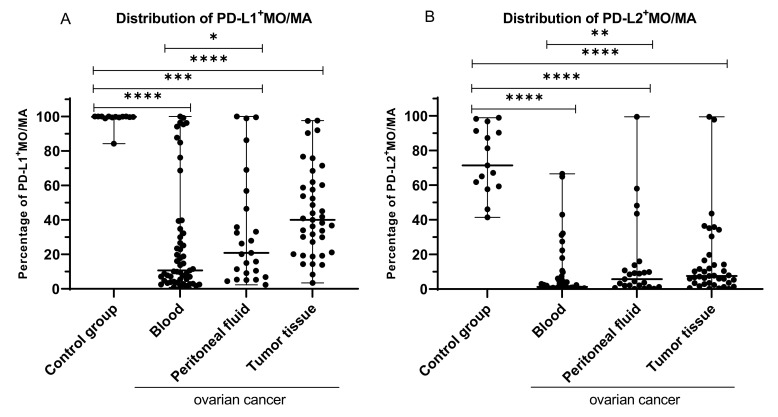
The difference in the distribution of PD-L1 (**A**) and PD-L2 positive MO/MA (**B**) in PB, PF, and TT of OC patients in comparison to healthy blood donors. The median value with the following marks indicate statistically significant differences: * *p* < 0.05, ** *p* < 0.01, *** *p* < 0.001, **** *p* < 0.0001.

**Figure 8 ijms-22-11563-f008:**
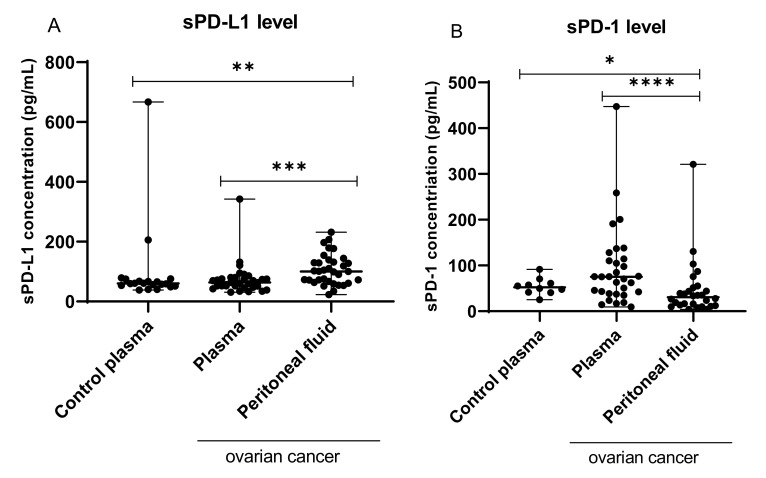
Levels of sPD-L1 (**A**; pg/mL) and sPD-1 (**B**; pg/mL) in the plasma of healthy blood donors, and the plasm and PF of patients with ovarian cancer. The median value with the following marks indicate statistically significant differences: * *p* < 0.05, ** *p* < 0.01, *** *p* < 0.001, **** *p* < 0.0001.

**Figure 9 ijms-22-11563-f009:**
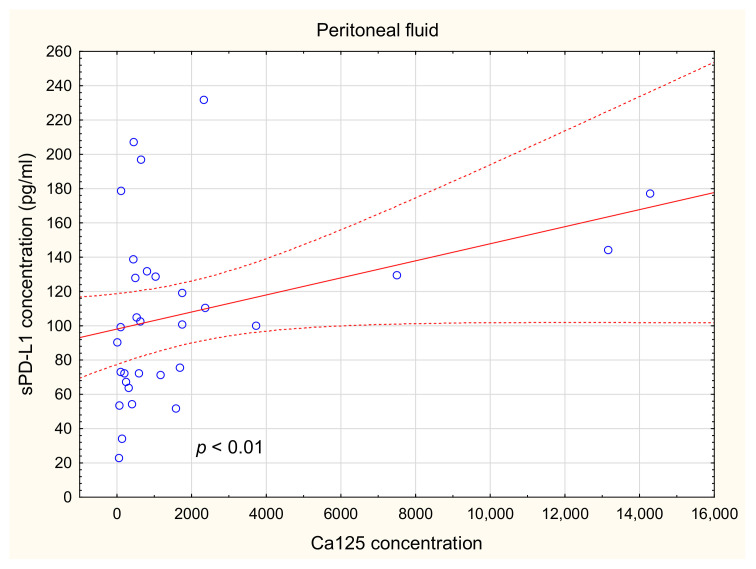
Relationship between the level of sPD-L1 in PF and Ca125 concentration in OC patients.

**Figure 10 ijms-22-11563-f010:**
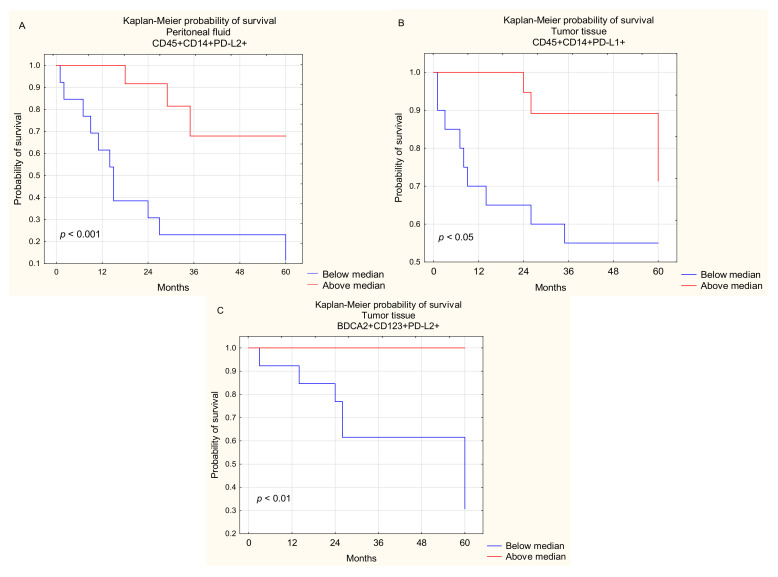
Relationship between CD45^+^CD14^+^PD-L2^+^ cells in PF (**A**), CD45^+^CD14^+^PD-L1^+^ cells in TT (**B**), and pDCs with PD-L2 expression in TT (**C**) and 5-year survival of OC patients.

**Figure 11 ijms-22-11563-f011:**
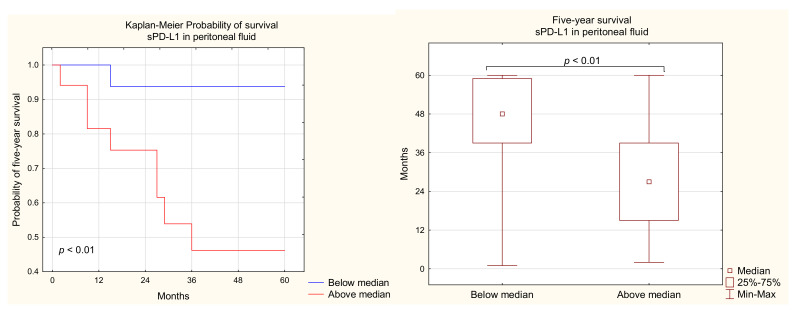
Relationship between sPD-L1 levels in the PF and 5-year survival of OC patients.

**Figure 12 ijms-22-11563-f012:**
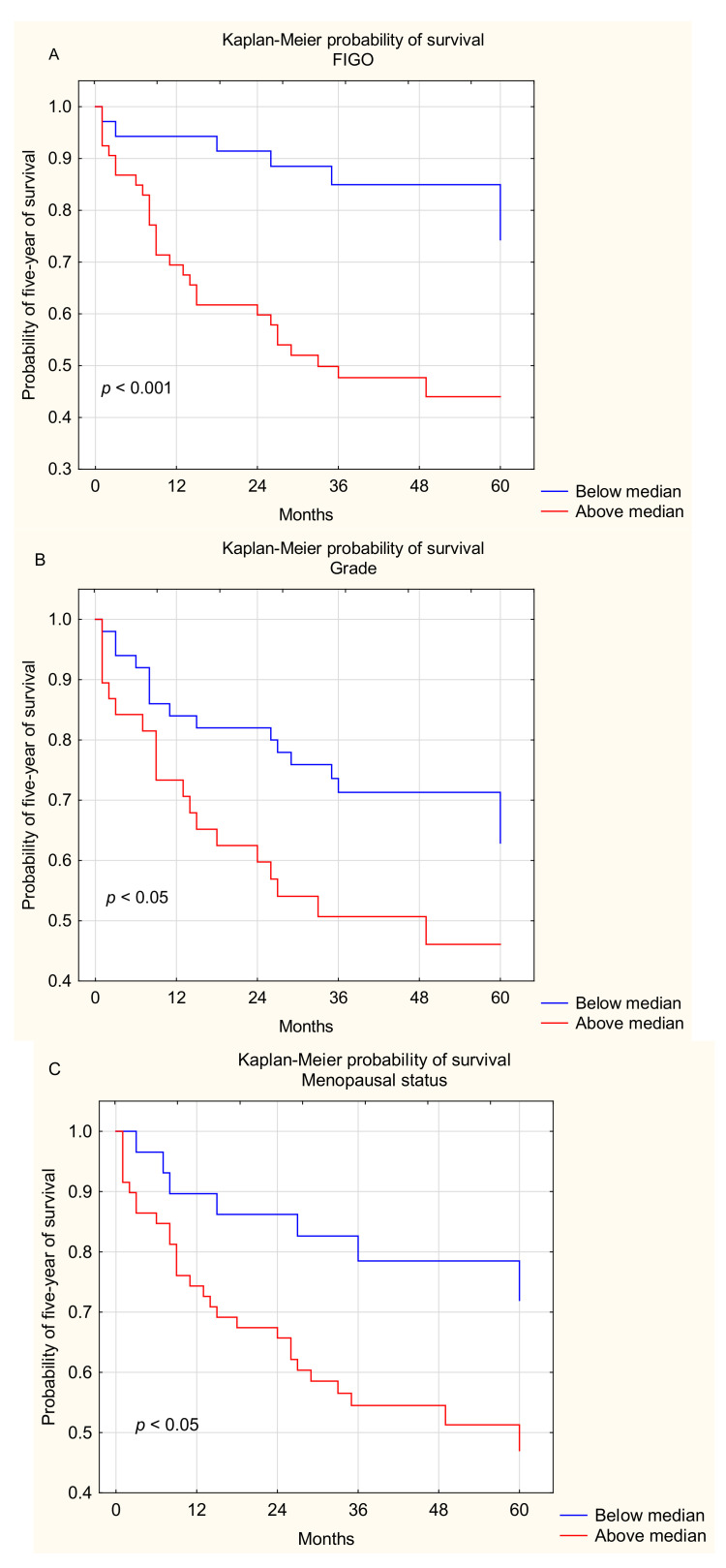
Five-year survival of OC patients with different FIGO stages (**A**), grades (**B**), and menopausal status (**C**).

**Figure 13 ijms-22-11563-f013:**
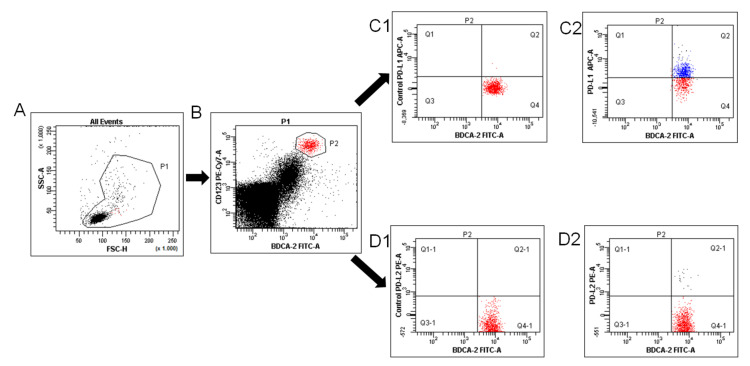
Flow cytometric analysis of BDCA-2^+^CD123^+^PD-L1^+^ and BDCA-2^+^CD123^+^PD-L2^+^ dendritic cells in the PF of ovarian cancer patients. An acquisition gate was established based on FSC and SSC that included mononuclear cells (P1 population; (**A**)). Next, the P1 gated MNCs were analyzed for BDCA-2^+^CD123^+^ (P2 population; (**B**)). The final dot plots of PD-L1 pDCs (region Q2, (**C2**)) or PD-L2 positive BDCA-2^+^CD123^+^ cells (region Q2-1, (**D2**)). Fluorescence minus one (FMO) control ((**C1**,**D1**), respectively) was used to verify the staining specificity and as a guide for setting the markers to delineate positive populations.

**Table 1 ijms-22-11563-t001:** Percentage of MO/MA and MO/MA with PD-L1 or PD-L2 expression in peripheral blood, peritoneal fluid, and among ovarian cancer infiltrating cells.

	Peripheral Blood			Peritoneal Fluid			Tumor Tissue		
	Median	Minimum	Maximum	Median	Minimum	Maximum	Median	Minimum	Maximum
%CD45^+^CD14^+^	23.65 °°^,^°°°	0.06	52.34	12.06	0.43	63.97	9.85	0.26	79.33
%CD45^+^CD14^+^PD-L1^+^	10.41	0.84	99.95	20.50	2.31	100.00	40.43 ****	3.44	97.59
%CD45^+^CD14^+^ PD-L2^+^	1.43	0.07	66.54	7.12 *	0.16	99.43	7.39 **	0.12	99.39

The median value with the following marks indicates statistically significant differences: * *p* < 0.05, ** *p* < 0.01, **** *p* < 0.0001 in relation to PB and °° *p* < 0.01 in relation to PF, °°° *p* < 0.001 in relation to TT.

**Table 2 ijms-22-11563-t002:** Levels of sPD-L1 (pg/mL) in the plasma and PF of patients with different FIGO stages, grades, and Kurman–Shih types of ovarian cancer.

Concentration of sPD-L1 (pg/mL) in Ovarian Cancer Patients		Plasma (*n* = 34)				Peritoneal Fluid (*n* = 34)		
		Median	Minimum	Maximum		Median	Minimum	Maximum
FIGO stage	I-II (*n* = 11)	69.70	33.15	131.81	I-II (*n* = 11)	75.54	61.25	207.27
	III-IV (*n* = 23)	59.54	31.32	342,31	III-IV (*n* = 23)	100.89	22.91	231.75
Grade	G2 (*n* = 18)	66.75	33.15	342.31	G2 *(n* = 18)	74.29	51.80	231.75
	G3 (*n* = 16)	61.25	31.32	131.81	G3 (*n* = 16)	110.85	22.91	207.27
Kurman and Shih OC Type	I (*n* = 25)	63.80	31.32	342.31	I (*n* = 25)	99.28	34.06	231.75
	II (*n* = 9)	59.54	34.06	131.81	II (*n* = 9)	128.68	22.91	207.27
Control Group (*n* = 20)	-	60.40	38.58	667.2		-	-	-

**Table 3 ijms-22-11563-t003:** Levels of sPD-L1 and sPD-1 (pg/mL) in the plasma and PF of OC patients and in the plasma of healthy donors.

	Ovarian Cancer Patients						Healthy Donors		
	Median	Minimum	Maximum	Median	Minimum	Maximum	Median	Minimum	Maximum
Concentration of sPD-L1 (pg/mL)	Plasma (*n* = 34)			PF (*n* = 34)			Plasma (*n* = 20)		
	63.37 *	31.32	342.3	100.5 °°°	22.91	231.8	60.40	38.58	667.2
Concentration of sPD-1 (pg/mL)	Plasma (*n* = 30)			PF (*n* = 27)			Plasma (*n* = 10)		
	72.91 ****	9.48	447.29	33.37	3.79	320.88	52.37 °	25.02	91.28

The median value with the following marks indicate statistically significant differences between the median value of the sPD-L1/sPD-1 concentration in the plasma and PF of the OC patients (* *p* < 0.05, **** *p* < 0.0001) and in the plasma of the OC patients and healthy blood donors (° *p* < 0.05, °°° *p* < 0.001).

**Table 4 ijms-22-11563-t004:** Clinical characteristics of ovarian cancer patients.

The Clinical Features	Ovarian Cancer Patients (*n* = 53)
Age (median), years (range)	55 (20–80)
FIGO Stage, *n* (%)	
Early (I–II)	22 (42%)
I	9 (17%)
II	13 (25%)
Advanced (III–IV)	31 (58%)
III	20 (37%)
IV	11 (21%)
The OC Classification According to Kurman and Shih, *n* (%)	
Type I (endometroid, serous G2, mucinous)	39 (74%)
Type II (serous G3)	14 (26%)
Grading (Histological Differentiation), *n* (%)	
Intermediate grade (G2)	29 (55%)
Low grade (G3)	24 (45%)
Menopausal Status	
Before menopause	21 (40%)
After menopause	32 (60%)
Ca125 range, median (U/mL)	525.30 (9.60–14283.0)
Body mass index (BMI) range, median	25.62 (18.34–38.37)

## Data Availability

All data generated or analyzed during this study are included in this publication.

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
