# Peer review of "Clinical and Prognostic Value of Antigen-Presenting Cells with PD-L1/PD-L2 Expression in Ovarian Cancer Patients"

_ijms, 2021, doi:10.3390/ijms222111563_

Round 1

Reviewer 1 Report

The authors conducted an important clinical study to understand the prognostic value of PD-L1/PD-L2 expression antigen-presenting cells in ovarian cancer patients. The study is based on a large patient number, therefore I would suggest publishing this paper in ijms if the author can address my following concerns.

Overall the introduction and discussion are well written, however, the results flow is very hard to follow. Data points are mistakenly plotted in multiple figures which has a very detrimental effect on the manuscript quality.

Major issue:

No proper conclusions are made based on the results. In the results section, the authors describe the results in a very repetitive style that makes the results section extremely hard to follow.

The data in Figure 2A control group are identical to Figure 4A control group, and the data in Figure 2A blood are spitted into two groups: FIGOI-II and FIGO III-IV. It is not OK to use the data twice if it is not explained in the text. And Figure 2B is partly replotted again in Figure 4B, the control group in Figure 4A and 4B are identical. Figure 4 C and 4D and Figure 3 control group results are identical and some of the data points in Figure 4C and Figure 4D FGO I-II and FIGO III-IV group are very likely identical as well.  Again, some of the data points in Figure 2 A, 2B and 3 are replotted in different ways in Figure 4 E F G I J K.

And Figure 5 and Figure 6 are very obscure, the data are barely mentioned in the main text, if there is no significant result find, it can be supplementary figures instead of the main figure. 

Again the data in Figure 11 are replotted in Figure 8, Figure 9 and Figure 10 based on different tumour stage criteria. I would suggest moving figures 8, 9 and 10 into supplementary if there is no significant difference found. It should be clearly stated in the main text that Figure 8, 9 and 10 are replotting the data in Figure 11 in order to show the effect of different stages of cancer on the number of specific immune cells.

Minor issues:

Different font sizes in the same figure, e.g. title font size for Figure 2 A and Figure 2 B are different, please check the font size, and align figure labelling.

Reviewer 2 Report

The present study well investigated the potential role of PD-L1/PD-L2 expression on antigen presenting cells and thus sPD-L1/sPD-1 as prognostic factor in ovarian cancer pathogenesis.

The manuscript is comprehensively written and well organized. However I suggest the authors to insert the therapy after surgery and a relative comment to better clarify the prognosis of the cohort along the time.

Round 2

Reviewer 1 Report

I agree to publish the revised manuscript in IJMS.